# Bone Microstructural Deterioration and miR-155/*RHOA*-Mediated Osteoclastogenesis in Type 2 Diabetes Mellitus

**DOI:** 10.3390/ijms26178159

**Published:** 2025-08-22

**Authors:** Mouza M. Alaleeli, Suneesh Kaimala, Ernest Adeghate, Sahar Mohsin

**Affiliations:** Department of Anatomy, College of Medicine and Health Sciences, United Arab Emirates University, Al-Ain P.O. Box 15551, United Arab Emirates

**Keywords:** type 2 diabetes mellitus, bone resorption, osteoclastogenesis, remodeling cycle

## Abstract

Type 2 diabetes mellitus (T2DM) is known to increase the risk of fragility fractures; however, the underlying mechanism is still elusive. Reduced miR-155 and elevated *RHOA* are known to drive bone resorption, but their role in T2DM remains unclear. This study investigates bone remodeling imbalances in T2DM through miR-155 and *RHOA* expression profiling. Three-month-old female Wistar rats were fed a high-calorie diet for 3 weeks, followed by intraperitoneal injections of two lower doses of streptozotocin at weekly intervals to induce T2DM. Bone analysis from diabetic rats tested using qRT-PCR showed significantly reduced miR-155 levels and elevated *RHOA.* Histological analysis showed a 12.65% increase in Tb.Sp, 10.07% decrease in Tb.Th, and significant increase (*p* < 0.05) in apoptotic osteocytes. The bone turnover marker CTx-1 level was increased by 20.84%, and RANKL levels were significantly increased in T2DM. IL-1β and TNF-α were increased in T2DM. Bone resorption is more likely to occur in T2DM as both IL-1β and TNF-α work synergistically to promote osteoclastogenesis. MiR-155 could be an important modulator of bone remodeling in T2DM and a potential therapeutic target for diabetic osteopathy.

## 1. Introduction

Type II diabetes mellitus (T2DM) is a metabolic disorder that is characterized by chronic hyperglycemia, which is caused by insulin resistance along with defective insulin secretion [1,2]. Insulin resistance is commonly attributed to aging, poor lifestyle habits, and obesity [2]. Globally, T2DM is on the rise, and the United Arab Emirates has one of the highest prevalences of T2DM and is ranked fifth among the Arab world countries [3,4]. Osteoporosis is one of the many health complications associated with T2DM [5]. Osteoporosis is a bone disease characterized by reduced bone mass and deteriorated bone quality, which elevates the likelihood of hip fractures [6]. Bone quantity refers to the assessment of bone mineral density, whereas bone quality encompasses the evaluation of bone microstructure [7,8]. A study by Mohsin et al. (2019) demonstrated that type 2 diabetes mellitus adversely impacts the trabecular architecture in the femoral head of rats [9]. An increase in apoptotic osteocytes is linked to type 2 diabetes mellitus (T2DM) [10]. This rise typically signals disrupted bone homeostasis, often associated with conditions that favor bone resorption or inhibit bone formation [11]. Apoptotic osteocytes may be driven by inflammatory cytokines like TNF-α (tumor necrosis factor-alpha) and IL-1β (Interleukin-1 beta), which contribute to osteocyte death and accelerated bone loss [12]. These cytokines promote bone resorption [12,13] by activating osteoclasts through TNFR and IL-1R receptors, triggering signaling pathways that enhance osteoclast differentiation [14]. Additionally, TNF-α and IL-1β increase RANKL expression in osteoblasts, lower OPG levels, and stimulate other inflammatory mediators that further boost osteoclast activity [14]. An imbalance between bone resorption and bone formation causes bone loss, which leads to thinning and perforation of the bone structure, ultimately changing its architecture and reducing its strength [15].

While T2DM-induced bone fragility has been linked to chronic inflammation and altered bone remodeling [16], the molecular mechanisms underlying these changes remain poorly understood. In particular, the role of microRNA-155 (miR-155) and its downstream effector *RHOA* in modulating osteoclastogenesis under diabetic conditions is not fully established. miR-155 has been implicated in the regulation of osteoclast differentiation, and *RHOA* is known to control actin cytoskeletal dynamics critical for osteoclast function [17,18]. However, it is unclear whether the dysregulation of miR-155 and *RHOA* contributes to the skeletal complications observed in T2DM. Understanding this regulatory axis could reveal novel mechanisms driving bone resorption and trabecular deterioration in diabetic bone disease.

Hence, the downstream signaling that drives osteoclastogenesis, particularly the RANK/RANKL/OPG pathway, may be influenced by inflammatory signals and miRNA-mediated regulation [19,20,21]. Exploring how miR-155 and *RHOA* interact with this key axis of bone remodeling in the context of diabetes could offer deeper insights into skeletal fragility associated with T2DM.

MicroRNAs can regulate gene expression by complementary base pairing to a target gene [22]. Research has shown that miR-155 deficiency can promote osteoclastogenesis [17]. *RHOA* (Ras homolog family member A), a molecular switch that codes for Rho subfamily GTPases that are important for regulating the actin cytoskeleton, could also promote osteoclastogenesis [18]. A study has shown that *RHOA* could be a possible target for miR-155 [23]. The bone resorption and bone formation rate is managed through the RANK/RANKL/OPG pathway [19]. RANKL (receptor activator of NF-kB ligand) is secreted by osteoblasts, the cells that are responsible for bone formation, while its receptor, RANK (receptor activator of NF-kB), is expressed by osteoclasts, the cells that are responsible for bone resorption [19]. The binding of RANKL with RANK triggers osteoclastogenesis, leading to increased bone resorption [19]. Osteoprotegerin (OPG), a decoy receptor that is also expressed by osteoblasts, competes with RANK for binding to RANKL, thereby inhibiting the RANKL-RANK interaction and preventing osteoclastogenesis [19]. Higher levels of RANKL are associated with increased bone resorption [24]. A high OPG/RANKL ratio promotes bone resorption [19,25].

T2DM is considered a chronic inflammatory disease, characterized by increased circulating proinflammatory cytokines [16]. Tumor necrosis factor-alpha (TNF-α) can alter insulin sensitivity, and IL-1β can cause pancreatic β-cell damage [16,26]. IL-1β is a key player in elevating bone resorption [20]. It is a proinflammatory cytokine produced by innate immune cells such as macrophages [27]. IL-1β binds to IL-1RI receptors on osteoclasts and stimulates osteoclastogenesis [20]. The proinflammatory cytokine tumor necrosis factor-alpha (TNF-α) works synergistically with IL-1β to stimulate this process [21].

In type 2 diabetes, it remains controversial whether bone resorption is enhanced or suppressed [28,29,30,31,32,33,34]. Some studies have reported elevated resorption rates in individuals with type 2 diabetes, while others have observed a decrease in bone resorption [31,34]. The conflicting findings regarding bone resorption in individuals with T2DM can be attributed to several contributing factors. Heterogeneity in patient populations including age, sex, ethnicity, duration of diabetes, and the presence of comorbidities can significantly influence bone turnover markers [21,28,29,30], and earlier studies have highlighted ethnicity-related differences in these markers [30]. Glycemic control also plays a pivotal role; poorly controlled diabetes is often associated with increased systemic inflammation and oxidative stress, which can enhance osteoclast activity and bone resorption, while well-controlled diabetes may be associated with suppressed bone remodeling [21,31]. Variations in methodologies such as the use of serum versus urinary resorption markers [28,32], imaging versus histological techniques, and differences in study designs further contribute to inconsistent findings across the literature. Additionally, the use of different experimental animal models of T2DM (e.g., genetic, diet-induced, or chemically induced models) may account for disparities in outcomes related to bone metabolism. Therefore, the current research is focused on determining the underlying mechanisms involved in altering the bone remodeling cycle, with a special focus on osteoclast function. In particular, we explore the role of miR-155, a potential molecular regulator of osteoclastogenesis, to help clarify the signaling pathways contributing to altered bone resorption in T2DM. Our findings aim to provide mechanistic insight that may help reconcile the discrepancies observed in previous studies.

This research seeks to resolve these discrepancies by measuring markers of bone resorption and investigating the role of miR-155 in T2DM. Specifically, it examines the expression profile of miR-155 in the bones obtained from rats with T2DM from our earlier study [9] to gain a deeper understanding of the mechanisms contributing to skeletal fragility in this condition. Additionally, this study analyses diabetes-induced microstructural changes in trabecular bone.

## 2. Results

### 2.1. Histological Analysis

#### 2.1.1. Masson’s Trichrome Staining for Trabecular Measurements

Paraffin sections of the proximal head of the femur from the control and T2DM animal groups were stained with Masson’s Trichrome to measure trabecular separation (Tb.Sp) and trabecular thickness (Tb.Th). As illustrated in Figure 1b, the trabecular separation (Tb.Sp), as shown by the green arrow in the T2DM group, was increased significantly (*p* = 0.00026) by 12.65%, while trabecular thickness (Tb.Th), shown by the yellow arrow, decreased by 10.07%, although this change was not statistically significant (*p* = 0.18656) compared to the control group. These changes may reflect compromised bone quality in T2DM, as they suggest increased bone porosity.

#### 2.1.2. TUNEL Staining for the Estimation of Apoptosis

Paraffin sections of the proximal head of the femur from the control and T2DM animal groups were examined using the TUNEL assay. A distinctly high number of apoptotic osteocytes were found in the T2DM animal group compared to the control animal group, as shown in Figure 2. This finding may indicate impaired bone remodeling, potentially associated with increased bone resorption in T2DM.

### 2.2. Estimation of Proinflammatory Cytokines

#### Significant Increase in IL-1β and Non-Significant Elevation in TNF-α in T2DM Rat Bones, Suggesting Elevated Inflammatory Activity

The protein expression levels of IL-1β and TNF-α were quantified from the tibia bone lysates of the control and T2DM groups by Western blotting, and normalized against GAPDH to plot their relative expression. There was a significant increase in IL-1β expression (*p* = 0.0010) and a non-significant (71.86%) increase in the expression of TNF-α (*p* = 0.2654) in diabetic rat bones, compared to the control, as illustrated in Figure 3a–c.

The significant rise in IL-1β and the upward trend in TNF-α suggest enhanced inflammatory signaling in the bone microenvironment of T2DM rats. IL-1β is a key mediator in osteoclast activation and bone resorption, and its upregulation may contribute to the observed bone deterioration. The increase in TNF-α, although not statistically significant, may still play a supporting role in promoting osteoclastogenesis and impairing bone formation in the diabetic condition. Together, these findings reflect a proinflammatory milieu in T2DM bones, potentially contributing to the imbalance in bone remodeling and reduced bone quality.

### 2.3. Significant Increase in RANKL and Non-Significant Elevation in CTx1, Suggesting Enhanced Bone Resorption in T2DM Rats

Western blot analysis was conducted to assess RANKL protein levels in bone lysates from both control and T2DM groups (see RANKL bands in Figure 3c). A significant upregulation of RANKL expression was observed in the bones of rats from the T2DM group (*p* = 0.007), as illustrated in Figure 4. This may reflect increased bone resorption, given RANKL’s role as a marker of osteoclast activity.

The Cross-Linked C-telopeptide of type I collagen (CTx1) is a bone resorption marker; it is secreted by the osteoclasts during bone resorption [35]. The CTx1 level in the bones of T2DM rats was raised (*p* = 0.352) by 20.84% compared to the control, indicating elevated bone resorption in the T2DM animal group. However, this result was statistically non-significant, as shown in Figure 5.

### 2.4. Profile Expression of MicroRNA-155 and Its Target Gene

#### 2.4.1. Significant Downregulation of miR-155 Expression in Bones of T2DM Rats Compared to Controls

Quantitative RT-PCR was used to measure miR-155 expression levels in bone samples from both control and T2DM groups, with normalization to U6 small nucleolar RNA for relative expression analysis. As shown in Figure 6, miR-155 expression was significantly reduced (*p* = 0.0179) in the bones of the T2DM group compared to the control group, suggesting enhanced osteoclastogenesis.

#### 2.4.2. Significant Upregulation of RHOA Gene Expression in Bones of T2DM Rats Correlates with Downregulated miR-155

According to the microRNA–target interaction database (miRTarBase), the *RHOA* gene has been identified as a potential target of miR-155 (ID: MIRT000949) [36]. To explore this interaction, the expression of *RHOA* was examined in T2DM bone samples, where miR-155 was found to be downregulated. Quantitative RT-PCR was used to measure *RHOA* levels, with normalization against β-actin for relative expression analysis. As illustrated in Figure 7, *RHOA* expression was significantly higher (*p* = 0.047) in the T2DM group compared to the control group. This finding aligns with the downregulated expression of miR-155, indicating enhanced osteoclastogenesis.

## 3. Discussion

Osteoporosis is a disorder that accelerates the levels of bone fragility and deteriorates the microarchitecture of the bone tissue, which leads to fracture susceptibility, clinically determined by measuring decreased levels of bone mineral density [37]. Most elderly people encounter osteoporosis in their lifetime, but studies have shown that this risk is increased in patients with type 2 diabetes predisposing them to increased fracture risk [38,39]. Some studies indicated that type 2 diabetes patients show high bone resorption rates, while others suggested a reduction in bone resorption [24,25]. Therefore, the current research is focused on determining the underlying mechanism involved in altering the bone remodeling cycle with a special focus on osteoclast function.

RNA interference is one of several mechanisms regulating gene expression, mediated by small endogenous non-coding RNA molecules known as microRNAs [40]. MicroRNAs function by binding to target gene mRNAs, leading to suppression of gene activity [40]. Consequently, elevated microRNA levels reduce gene expression, whereas lower microRNA levels enhance it [40]. Numerous studies have demonstrated that changes in microRNA expression are associated with various diseases, suggesting their potential use as diagnostic biomarkers [40]. However, it is still unknown whether increased bone resorption is a feature of T2DM. MiR-155 and *RHOA* both play a role in osteoclastogenesis [17,18], and the miRTarBase database showed that *RHOA* could be a possible target for miR-155 (ID: MIRT000949) [36]. This was evident in a knockdown study and an overexpression study of miR-155, and the studies stated that overexpression of miR-155 downregulates *RHOA* and knockdown of miR-155 upregulates *RHOA* [23]. To further confirm this, the study performed the *RHOA* gene 3′ UTR luciferase reporter assay, which reported that the luciferase activity of *RHOA* was reduced when miR-155 was expressed, which means that RHOA could be a possible target for miR-155 [23]. Studies confirmed that miR-155 is regulated by targeting *RHOA* in epithelial cells [23,41]. This study is novel as the link between miR-155 and *RHOA* has not been shown before in T2DM diabetic osteopathy.

Increased bone porosity, thinning of trabeculae, and reduced bone volume and BMD are features of osteoporosis [42,43,44]. In this study, we observed reduced bone quality in T2DM, as trabecular separation increased by 12.65% and trabecular thickness decreased by 10.07% compared to the control group. These findings align with our previous study in which we investigated changes in the trabecular bone microstructure within the femoral head using micro-CT, which also showed statistically significant increased trabecular separation in the bones of T2DM rats (*p* < 0.05), reduced BV/TV, and reduced BMD (*p* < 0.05) [9], hence increasing the bone fragility and risk of fractures in T2DM.

An increased number of apoptotic osteocytes is associated with T2DM [10]. A study revealed that an increase in apoptotic osteocytes in bone typically indicates a disruption in bone homeostasis, often associated with conditions that promote bone resorption or reduce bone formation [11]. Elevated apoptotic osteocytes are linked to inflammatory conditions, where cytokines such as TNF-α (tumor necrosis factor-alpha) and IL-1β (Interleukin-1 beta) contribute to osteocyte death, leading to accelerated bone loss [45]. Proinflammatory cytokines, TNF-α and IL-1β, play a key role in promoting bone resorption [12,13]. They directly trigger the formation and activation of osteoclasts by interacting with osteoclast precursors and mature osteoclasts via TNFR (TNF receptor) and IL-1R (IL-1 receptor), activating signaling pathways that promote osteoclast differentiation [14]. Steeve et al. have shown that TNF-α and IL-1β increase RANKL expression in osteoblasts and reduce OPG levels, while also stimulating additional inflammatory mediators that enhance osteoclast activity [14]. In our analysis, the apoptotic osteocytes were raised in number and the expression levels of TNF-α and IL-1β were increased in the T2DM rats, which likely contributes to the increased bone resorption observed in type 2 diabetes.

To further support this finding, levels of C-terminal telopeptide of type I collagen (CTx1)—a marker released during osteoclast-mediated bone resorption [35]—were measured. Our results showed a 20.84% increase in CTx1 levels in the bones of T2DM rats, reinforcing the evidence of enhanced bone resorption in the diabetic group. We further investigated the mechanism by examining RANKL, a bone resorption marker that promotes osteoclast formation (osteoclastogenesis). In our study, the upregulation of RANKL expression in T2DM rats points to elevated bone resorption in the T2DM group. 

In previous studies, reduced expression of miR-155 has been associated with increased osteoclastogenesis and bone resorption [11,46]. For instance, research on orthodontic patients found that lower levels of miR-155 correlated with enhanced osteoclast differentiation and more severe root resorption [46]. In our study, bones from T2DM rats exhibited significantly decreased miR-155 expression, accompanied by an upregulation of *RHOA*. *RHOA* is a key regulator of the actin cytoskeleton, essential for podosome assembly and osteoclast mobility, and has been shown to promote osteoclastogenesis [18]. These findings support the likelihood of enhanced bone resorption in T2DM. The context-dependent role of miR-155, which may inhibit osteoblast function in some settings while suppressing osteoclastogenesis in others, highlights the complex regulation of bone remodeling in T2DM and suggests that reduced miR-155 expression contributes to enhanced bone resorption under diabetic conditions.

To the best of our knowledge, this is the first report demonstrating an inverse expression pattern of miR-155 and *RHOA* in diabetic bone tissue. This observation, together with elevated inflammatory cytokines and osteoclast activation markers (RANKL, CTX-1), suggests that miR-155 may modulate osteoclastogenesis, potentially through *RHOA*-related signaling pathways. While the regulatory interaction may be indirect, our in vivo data provide novel insights and highlight the need for future studies to elucidate the mechanistic link between miR-155, *RHOA*, and bone resorption in the context of T2DM.

MicroRNAs are promising biomarkers for diagnosing type 2 diabetes mellitus since they regulate key processes in bone remodeling and remain stable in the body. Profiling these miRNAs could allow early detection of bone changes in T2DM patients before clinical symptoms develop. Moreover, targeting miRNA pathways may offer new therapeutic options to reduce bone fragility and improve bone health. Incorporating miRNA analysis into clinical practice could enhance personalized treatment and patient outcomes.

## 4. Materials and Methods

### 4.1. Animal Model

The study utilized bones from three-month-old female Wistar rats (n = 16), which had been stored at −80 °C following a previous investigation [9]. Three-month-old female Wistar rat bones were used as they represent skeletally mature young adults commonly used in bone metabolism studies. These samples were available from our previously approved and published study, enabling adherence to the 3Rs by reducing additional animal use [47,48]. These rats were sourced from the animal house facility at the United Arab Emirates University. During a two-week acclimatization period, the animals were individually housed in cages under standard conditions, including a 12 h alternating light and dark cycle (22–24 °C), 50–60% humidity, and free access to standard rat chow and water ad libitum. All necessary measures were taken to minimize animal suffering and reduce the overall number of animals used. The Animal Ethics Committee of the College of Medicine and Health Sciences of the United Arab Emirates University approved all the animal experiments (ERA_2017_5597). The experimental rats were randomly divided into two groups (n = 8 for each group): the first group is the “control” group and the second group is the “T2DM” group. T2DM rats were fed a high-calorie diet (D12492 diet; Research Diets, Inc., New Brunswick, NJ, USA) for 3 weeks, followed by the injection of two lower doses of streptozotocin (STZ) (30 mg/kg intraperitoneally), which was administered at weekly intervals [49,50]. Three days after the last injection, tail vein blood glucose was measured after fasting for 5 h using a blood glucose meter (Accu-Chek Performa; Roche Diagnostics, Indianapolis, IN, USA). Rats having blood glucose > 15 mmol/L were considered diabetic and were used for our study (to check the body weight and the blood glucose measurements of the rats, see Table 1) [9]. Insulin resistance in diabetic animals was further confirmed through an insulin tolerance test. After confirmation of diabetes, both the control and T2DM groups were maintained for an additional 10 weeks to allow chronic disease changes to develop before sacrifice. The femur and tibia were selected for this study due to their high susceptibility to fragility fractures [51].

### 4.2. Masson’s Trichrome Stain

The Trichrome stain Masson (Sigma-Aldrich, St. Louis, MO, USA; lot: SLBN7822V) was applied to paraffin sections with a thickness of 5 µm. The sections were deparaffinized and rehydrated with distilled water. Bouin’s solution, preheated to 56 °C, was added to cover the sections, which were incubated for 1 h and then washed with tap water for 30 min. After the yellow coloration was removed with tap water, hematoxylin was applied for 20 min. The sections were then rinsed in running tap water for 10 min, followed by a wash in distilled water. Acid fuchsin was applied and the sections incubated for 20 min before being rinsed in distilled water. Phosphomolybdic–phosphotungstic acid solution was then added and incubated for 20 min. Aniline blue was applied and incubated for 20 min. Subsequently, 1% acetic acid was added, and the sections were incubated for 8 min, rinsed in distilled water, and dipped in 95% ethanol for 1 min. The sections were dehydrated in 100% ethanol and cleared in 100% xylene. Finally, the stained sections were mounted with DPX and covered with a coverslip, after which they were examined under a brightfield microscope. Trabecular separation (Tb.Sp) and trabecular thickness (Tb.Th) were measured.

### 4.3. TUNEL Assay

The TUNEL assay (terminal deoxynucleotidyl transferase dUTP nick end labeling) was performed using the TUNEL Assay Kit-HRP-DAB (ab206386, Abcam, Cambridge, UK). This method employs terminal deoxynucleotidyl transferase (TdT) enzyme, which attaches to the exposed 3′-OH termini of fragmented DNA within paraffin-embedded bone tissue sections to identify apoptotic nuclei. After binding, TdT facilitates the incorporation of biotin-labeled nucleotides. The biotin is then detected by a streptavidin conjugated to horseradish peroxidase (HRP). Visualization of the HRP-labeled nuclei is achieved through a reaction with diamino-benzidine (DAB), producing a brown-colored stain. Femoral bones from control and T2DM groups were fixed, embedded in paraffin, and cut into 5 µm sections for the assay. Apoptotic and live cells were counted per section, and the average cell counts were calculated for comparative analysis.

### 4.4. Enzyme-Linked Immunosorbent Assay (ELISA)

Protein extraction from the control and T2DM animal groups was performed on the tibia bone. The ELISA kit utilized for Cross-Linked C-telopeptide of type I collagen [CTxI] (CEA665Ra, Cloud-Clone Corp., Wuhan, China ) employed a competitive inhibition enzyme immunoassay approach. The microplate wells of the kit were coated with a monoclonal antibody specific to CTx1. Bone samples (bone hydrolysate) and Avidin conjugated to horseradish peroxidase (HRP) were added to each well. The Avidin–HRP enzyme conjugate competed with CTx1 in the bone samples for the binding sites. The unbound CTx1 (either from the bone samples or the enzyme conjugate) was washed off. Once the TMB (3,3′,5,5′-Tetramethylbenzidine) substrate was added, the wells that exhibited a color change were the ones that contained CTx1. It was measured spectrophotometrically at a wavelength of 450 nm.

### 4.5. Western Blotting

The radioimmunoprecipitation assay (RIPA) buffer was used to homogenize the tibia bone obtained from the control and T2DM animal groups to create bone lysate with equal amounts of protein. These bone lysates were used to measure the expression level of RANKL, TNF-α, and IL-1β. The bone lysates were mixed with the loading buffer. The samples were loaded into SDS-PAGE (sodium do-decyl sulfate–polyacrylamide gel electrophoresis) gel to allow protein separation by mass. The bands that contained the proteins were transferred to the PVDF (poly-vinylidene difluoride) membrane. The membrane was incubated in a blocking buffer at 4 °C overnight. Then, the membrane was incubated with the primary antibodies specific to the protein of interest (Table 2) for 1 h at room temperature. The membrane was washed with TBS-T and incubated with horseradish peroxidase (HRP)-conjugated secondary antibodies (Table 2) for 1 h at room temperature. The membrane was rewashed with TBS-T. ECL Plus Chemiluminescence reagent was placed onto it to detect the HRP activities and normalized against GAPDH to plot their relative expression.

### 4.6. Quantitative Real-Time PCR

Total RNA was extracted from bone tissues of both control and T2DM groups using the mirVana™ miRNA Isolation Kit (Invitrogen, Thermo Fisher Scientific, Waltham, MA, USA, LOT:00802875). For reverse transcription, TaqMan^®^ Small RNA Assays (Applied Biosystems, Thermo Fisher Scientific, Waltham, MA, USA, LOT:00991753) were used for miR-155, while the High-Capacity cDNA Reverse Transcription Kit (Applied Biosystems, LOT:00996805) was used for *RHOA* mRNA. PCR amplification was carried out using the TaqMan^®^ Universal PCR Master Mix (LOT:2005167, Applied Biosystems). Relative expression was calculated using the 2^−ΔΔCt^ method. U6 snRNA and β-actin were used as internal controls for miR-155 and *RHOA*, respectively. All reactions were run in duplicate and repeated three times for consistency.

### 4.7. Statistical Analysis

All data are expressed as means ± SEM. Student’s *t*-test or one-way ANOVA was used to compare all data using GraphPad Prism 5. The statistical significance levels used in this study were * *p* < 0.05, ** *p* < 0.01, and * *p* < 0.001. For in vivo assays, each group included n = 3 biological replicates (total n = 6). For RT-qPCR, each biological RNA sample was analyzed in technical triplicates, and the average of the technical replicates was used for statistical analysis. In vitro experiments were independently repeated in duplicate.

## 5. Conclusions

Type 2 diabetes mellitus (T2DM) negatively impacts bone health by disrupting bone remodeling and promoting bone resorption. The bone resorption marker CTx-1 was elevated by 20.84%, and RANKL expression was significantly increased, both indicative of enhanced osteoclastic activity. The upregulation of proinflammatory cytokines IL-1β and TNF-α further supports active osteoclastogenesis. Structural deterioration of bone microarchitecture was evident, with a 12.65% increase in trabecular separation and a 10.07% decrease in trabecular thickness. Structural changes were confirmed by micro-CT analysis in the same cohort of animals as in our earlier study [9], reinforcing the consistency of the observed bone degradation. An increased number of apoptotic osteocytes further indicates an imbalance in bone homeostasis by triggering a cascade of events leading to enhanced activity of osteoclasts and reduced osteoblast activity.

Molecular analysis revealed a downregulation of miR-155 and upregulation of *RHOA*, both of which promote osteoclastogenesis. These findings suggest that miR-155 may act as a key regulator of bone remodeling in T2DM and could serve as a potential therapeutic target for diabetic osteopathy.

## 6. Limitation of the Study

While our results reveal an inverse relationship between miR-155 and *RHOA* expression in diabetic bone, this study does not confirm a direct regulatory interaction between the two. Further validation is needed to establish the mechanistic link and confirm whether *RHOA* is a direct downstream target of miR-155 in the context of diabetic bone remodeling. Additionally, although we observed elevated levels of proinflammatory cytokines (TNF-α and IL-1β) and discussed their role in osteoclastogenesis, the potential crosstalk between these cytokines and the miR-155/*RHOA* pathway was not investigated. Furthermore, the relatively small sample size used in this study may limit the generalizability of the findings.

Future studies should include functional validation using miR-155 knockdown or overexpression (e.g., mimics/inhibitors) and *RHOA* siRNA, along with luciferase reporter assays and cytokine modulation, to elucidate these regulatory dynamics more conclusively.

## Figures and Tables

**Figure 1 ijms-26-08159-f001:**
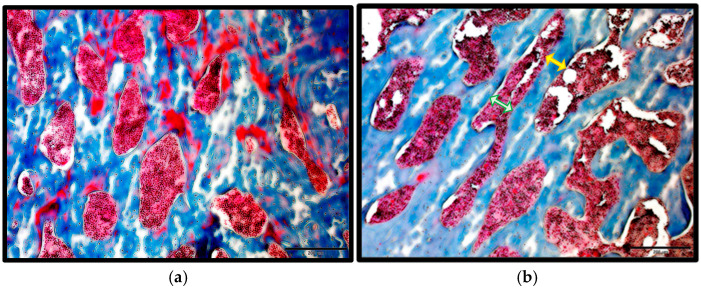
Masson’s Trichrome Staining. (**a**) Control group; (**b**) T2DM group. Scale bar = 200 μm. Trabecular separation (Tb.Sp) is shown by a green arrow, and trabecular thickness (Tb.Th) is shown by a yellow arrow. (**c**) Significant increase in Tb.SP, *** = *p* =< 0.001, in T2DM. (**d**) Non-significant decrease (*p* = 0.18656) in Tb.Th in T2DM.

**Figure 2 ijms-26-08159-f002:**
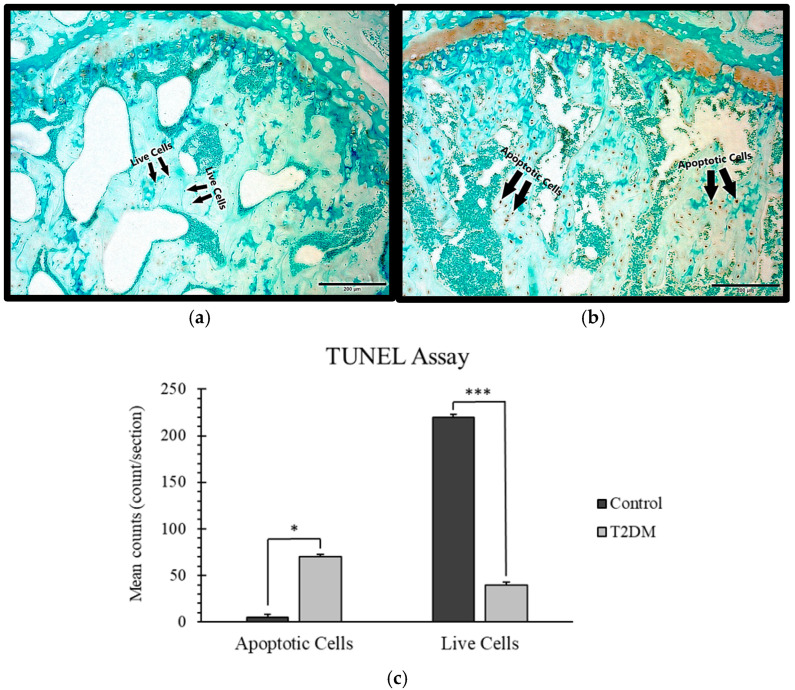
TUNEL assay demonstrating apoptotic osteocytes. (**a**) Control group, showing live osteocytes as blue dots; (**b**) T2DM group, showing apoptotic osteocytes as brown dots. Scale bar = 200 μm. (**c**) Mean counts of the apoptotic and live cells in the control and T2DM groups. * = *p* < 0.05, *** = *p* =< 0.001.

**Figure 3 ijms-26-08159-f003:**
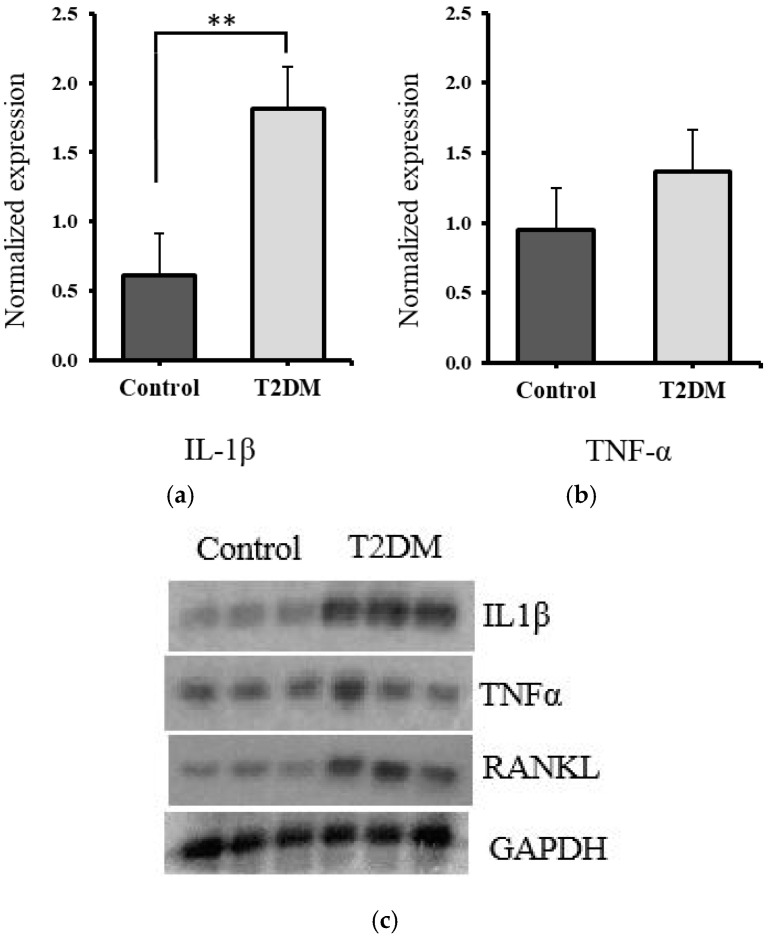
The expression of IL-1β and TNF-α in the bones of control and T2DM groups (n = 3/group). (**a**) IL-1 β expression is significantly elevated in the T2DM animal group (** *p* = 0.0010); (**b**) TNF-α expression is non-significantly increased in the T2DM animal group (*p* = 0.2654). (**c**) Representative Western blot showing IL-1β, TNF-α, and RANKL protein levels, with GAPDH as the loading control.

**Figure 4 ijms-26-08159-f004:**
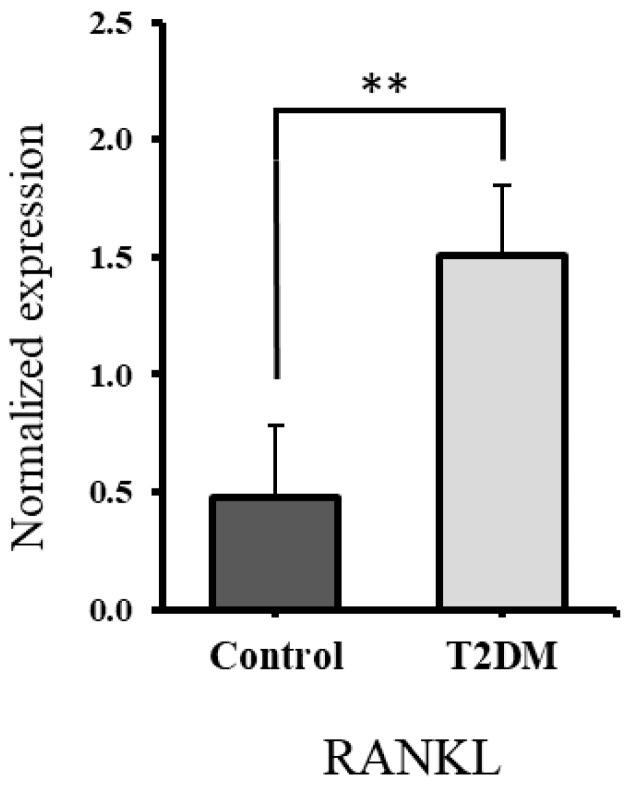
The expression of RANKL was increased in the bones of T2DM rats. RANKL expression is significantly upregulated in the T2DM animal group ** *p* = 0.007.

**Figure 5 ijms-26-08159-f005:**
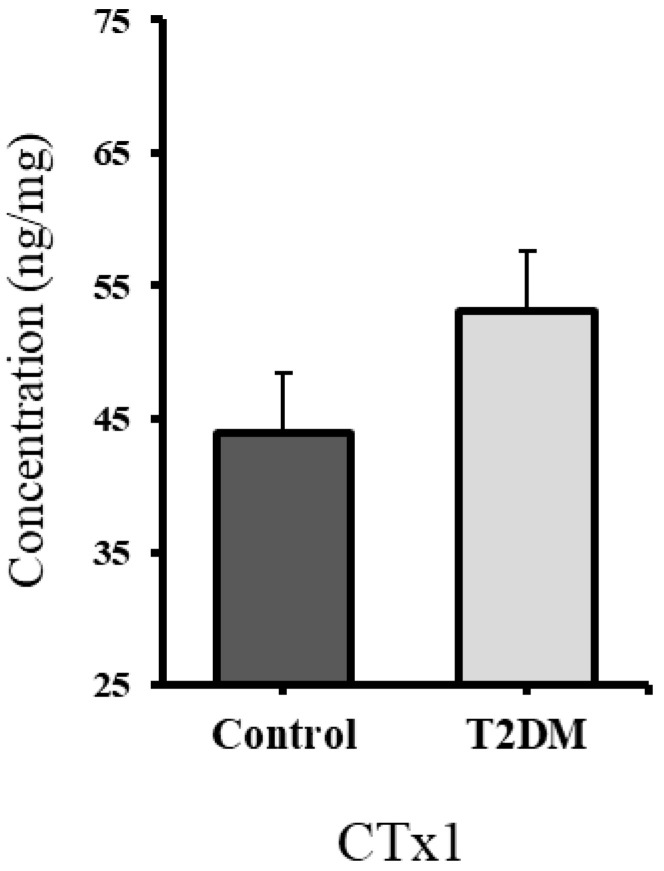
The CTx1 level in the bones of the T2DM animal group is increased non-significantly by 20.84%. *p* = 0.352.

**Figure 6 ijms-26-08159-f006:**
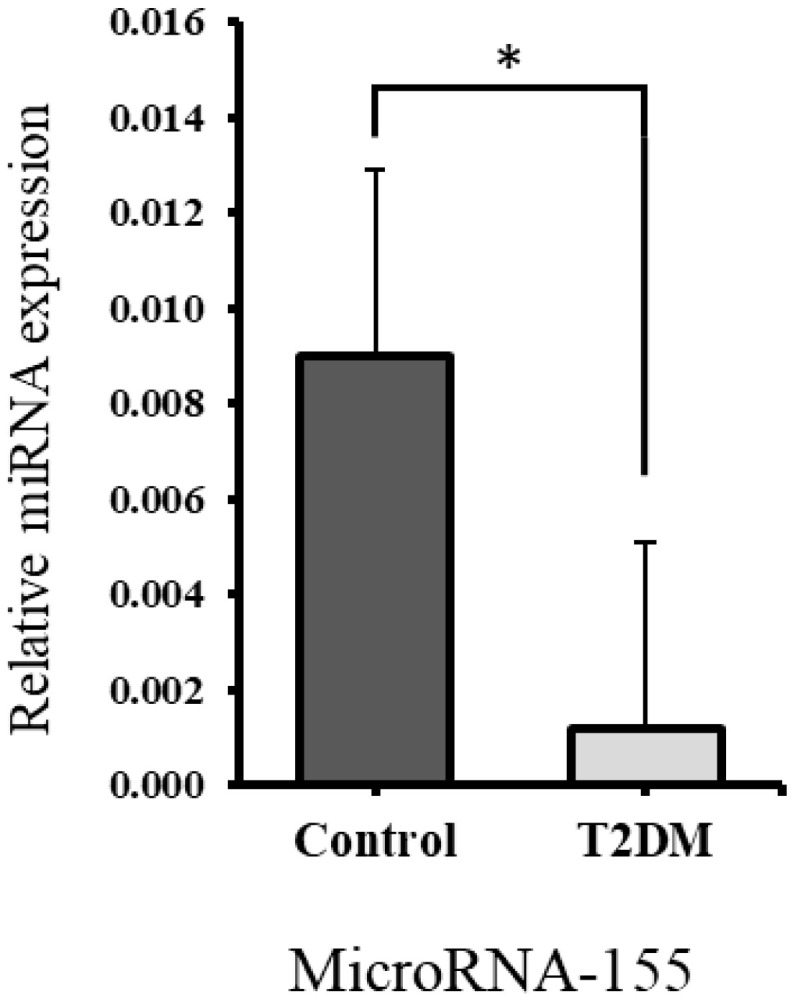
The expression of microRNA-155 was reduced significantly in the T2DM group * *p* = 0.0179.

**Figure 7 ijms-26-08159-f007:**
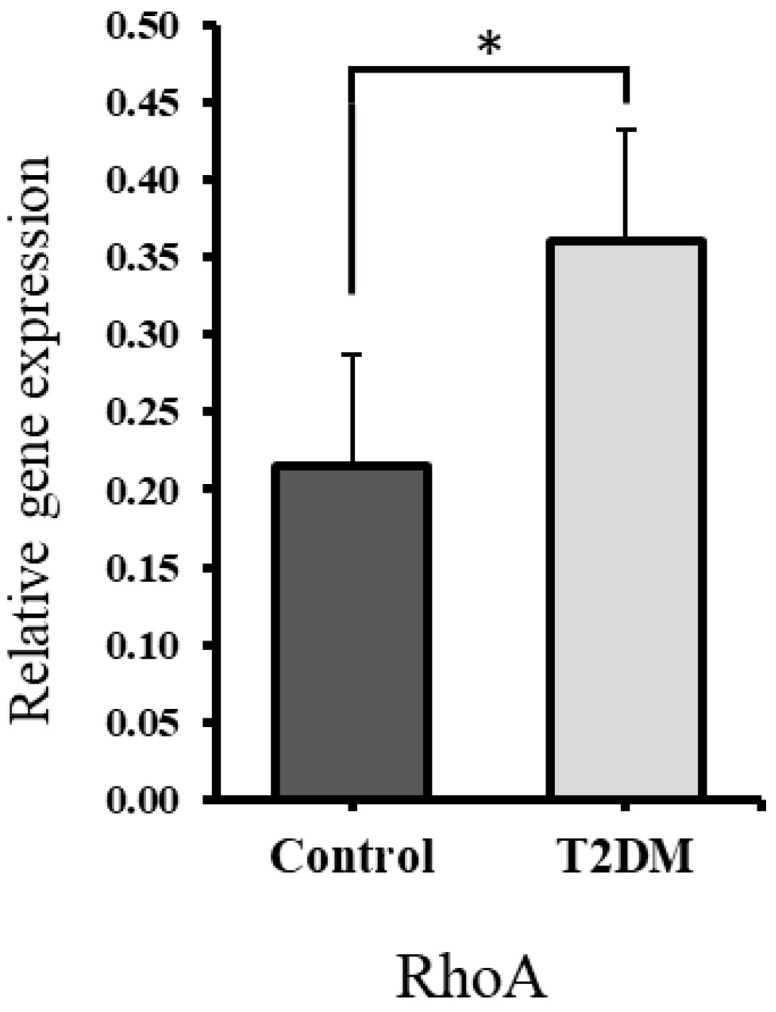
The expression of the *RHOA* gene was upregulated significantly * *p* = 0.047.

**Table 1 ijms-26-08159-t001:** Mean ± S.D blood glucose and body weight measurements [9].

	Blood Glucose (mmol/L)	Body Weight (gm)
Control	6.34 ± 0.46	243.6 ± 26.2
T2DM	24.50 ± 2.90	203.5 ± 25.98

**Table 2 ijms-26-08159-t002:** The proteins of interest and their primary and secondary antibodies.

Protein of Interest	Primary Antibodies	Secondary Antibodies
RANKL	Santa Cruz Biotechnology, Dallas, TX, USA Cat: Sc-377079	Abcam, Cambridge, UK Cat: ab 205719
TNF-α	Santa Cruz Biotechnology, Dallas, TX, USA Cat: Sc-52746	Abcam, Cambridge, UK Cat: ab 205719
IL-1β	Cloud-Clone Corp., Wuhan, China Cat: PAA563Ra01	Cell Signaling Technology, Danvers, MA, USA Cat: 7074S
GAPDH	Santa Cruz Biotechnology, Dallas, TX, USA Cat: Sc-32233	Abcam, Cambridge, UK Cat: ab 205719

## Data Availability

The data supporting the findings of this study are available within the paper. Should any raw data files be needed, they are available from the corresponding author upon reasonable request.

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
