# Peer review of "Bone Microstructural Deterioration and miR-155/RHOA-Mediated Osteoclastogenesis in Type 2 Diabetes Mellitus"

_ijms, 2025, doi:10.3390/ijms26178159_

Round 1
Reviewer 1 Report
Comments and Suggestions for Authors
In this study, Alaleeli and colleagues utilize animal models of type 2 diabetes mellitus (T2DM) to explore the causes of compromised bone structure. They report intriguing findings suggesting that bone health in T2DM may deteriorate due to increased bone resorption, associated with alterations in microRNA-155 (miR-155) and RhoA signaling. This study offers valuable insights from both basic science and clinical perspectives. However, additional experiments elucidating the underlying mechanisms would further strengthen the conclusions. Addressing the following concerns could enhance the clarity and impact of the presented findings for readers.
Major Points:
- The authors demonstrate that miR-155 and RhoA are key factors in regulating osteoclastogenesis and bone resorption in rat models of type 2 diabetes mellitus (T2DM). However, the study lacks direct evidence for a regulatory interaction between miR-155 and RhoA, representing a critical gap. Previous research suggests that miR-155 may indirectly influence osteoclastogenesis by modulating RhoA-related signaling pathways. It remains unclear whether the current study supports this concept.
- Previous studies have demonstrated that microRNA-155 (miR-155) exerts a catabolic effect on osteogenesis and bone mass, suggesting inhibition of miR-155 as a potential strategy for bone regeneration and bone defect healing. In contrast, the current study reports lower miR-155 expression in animals with type 2 diabetes mellitus (T2DM) exhibiting poorer bone quality. Can the authors provide a detailed explanation for this discrepancy?
- The authors examine the roles of pro-inflammatory cytokines (TNF-α and IL-1β) in promoting bone resorption, highlighting their synergistic effects in stimulating osteoclastogenesis. However, the study does not investigate potential interactions between these cytokines and the miR-155/RhoA pathway.
- Micro-CT scanning is an effective method for evaluating bone phenotype. This data is critical for the current study, and it is unclear whether the authors conducted relevant micro-CT experiments.
- The streptozotocin (STZ)-induced diabetic model in rodents is a widely used experimental model for type 1 diabetes mellitus (T1DM). It mimics the insulin deficiency and hyperglycemia characteristic of human T1DM by selectively destroying pancreatic beta cells. This study examines bone quality in type 2 diabetes mellitus (T2DM) but employs an animal model induced with type 1 diabetes, which is counterintuitive. To investigate bone quality in T2DM rats, using a high-fructose or high-fat diet to induce an experimental model would be more appropriate.
- In the Materials and Methods section, the authors report using 16 rats in the experiment. However, the Western blot data include results from only three rats (n=3) to evaluate inflammation status, which undermines the statistical robustness and persuasiveness of the evidence.
- In the PCR experiments, the miR-155 data from diabetic rats exhibit excessively large error bars, indicating high variability within the group. Although the authors report a statistically significant difference compared to the control group, this result is questionable due to the high variability. Furthermore, the authors’ attempt to characterize the sample size (n) is unclear, making it challenging to assess the reliability of the data.
The manuscript needs editing by a native English speaker.
Author Response
We thank the reviewer for their time and constructive feedback. The manuscript has now been carefully revised for language, grammar, and clarity by a native English speaker with scientific writing expertise. We have improved the sentence structure and overall readability. We hope the revised version meets the expected linguistic standards.
In the attachment, we have provided a point-by-point response to all reviewer comments, indicating the changes made in the revised manuscript.

Reviewer 2 Report
Comments and Suggestions for Authors
Although the authors indicate that MiR-155 could be an important modulator of bone remodeling in T2DM and a potential therapeutic target for diabetic osteopathy.However, there are still some limitations:
- The introduction attempts to cover a wide range of aspects related to T2DM and bone remodeling. However, it lacks a clear focus or rationale connecting all elements to the research objective. A sharper focus on the specific gaps in current knowledge about miR-155 and RhoA in the context of T2DM-induced bone remodeling is needed.
- The transition between discussing miR-155 and the broader bone remodeling mechanisms (like RANKL-RANK-OPG) is abrupt. Adding a connecting paragraph or sentence could improve the narrative flow.
- All the results are described too simply, lacking logical progression and simple summary.
- The changes in trabecular separation (Tb.Sp) and thickness (Tb.Th) are quantified, but the absence of statistical measures (e.g., p-values or confidence intervals) weakens the validity of the results. Adding these metrics would strengthen the findings.
- The 2.1.2 section lacks any discussion of the biological relevance or implications of increased osteocyte apoptosis in T2DM, which would help contextualize the findings.
- The p-value for TNF-α (p = 0.2654) indicates that the increase is not statistically significant, yet it is reported alongside IL-1β as a noteworthy observation. This could be misleading. Clarify that the increase in TNF-α expression is not statistically significant.
- Reference to Figures 4 and 5 lacks adequate detail. Provide descriptions of what these figures represent, including visual cues (e.g., bands for RANKL in Figure 4 and assay results for CTx1 in Figure 5).
- Figure 1.2 appears two times.
- The discussion mentions potential diagnostic applications of miRNAs but does not sufficiently link this to clinical outcomes or therapeutic implications for T2DM patients. Highlighting these aspects would enhance the study’s impact.
- The rationale for using three-month-old female Wistar rats is not provided. It is unclear why this age and sex were chosen over other potential models.
Author Response

(The authors gave the same response as above.)

Reviewer 3 Report
Comments and Suggestions for Authors
After reviewing this study, I think, to evaluate the hypothesis, the experimental design is complete. To improve the novelty and readability, I have several suggestions:
- The rationale of this study is "In type 2 diabetes, it remains controversial whether bone resorption is enhanced or suppressed. Some studies have reported elevated resorption rates in
individuals with type 2 diabetes, while others have observed a decrease in bone resorption. This research seeks to resolve these discrepancies by measuring markers of bone resorption and investigating the role of miR-155 in T2DM." Despite the authors stated a series of experiment to address this rationale, no discussion regarding the possible reason to this discrepancy is pity. I would suggest the authors discuss this issue in this study. - Some figures lack statistical examination, including Figure 1, 3b, and 5.
- For Figure 2, I suggest the authors grouping the data with "apoptotic cell" and "living cell", which would be more clear.
- For the subheading of result section, I suggest using a sentence of summary as the subheading.
Author Response
We thank the reviewer for their time and constructive feedback. In the attachment, we have provided a point-by-point response to all reviewer comments, indicating the changes made in the revised manuscript.

Round 2
Reviewer 2 Report
Comments and Suggestions for Authors
1.While downregulation of miR-155 and upregulation of RhoA were observed, a direct regulatory relationship between the two has not been demonstrated.Please increase the maintenance time of T2DM, such as continuous observation for 4–8 weeks after induction, to more realistically reflect the chronic changes of diabetic bone disease.
2.Although the method of “high-calorie diet for 3 weeks + 2 STZ injections” to induce T2DM is commonly used, it only takes a few weeks and may not be enough to simulate the long-term effects of chronic T2DM on bone metabolism.Please verify its function by using miR-155 mimic/inhibitor or RhoA siRNA.
Author Response
We sincerely thank the reviewer for their time and constructive feedback, which has greatly improved the clarity and quality of our manuscript. Additionally, the manuscript has been revised by a native English speaker to enhance the language and readability.
Comment 1
1.While downregulation of miR-155 and upregulation of RhoA were observed, a direct regulatory relationship between the two has not been demonstrated.Please increase the maintenance time of T2DM, such as continuous observation for 4–8 weeks after induction, to more realistically reflect the chronic changes of diabetic bone disease.
Response:
Thank you for your comment. We would like to clarify that in the present study, both control and T2DM groups were maintained for 10 weeks after the onset of diabetes before sacrifice, allowing sufficient time for chronic metabolic and skeletal changes to develop. This is already stated in the manuscript (page 11, first paragraph, line 397):
“Both groups were sacrificed at 10 weeks of the onset of diabetes.”
In our earlier study on the same cohort of animals (Mohsin et al., 2019), micro-CT analysis demonstrated significant deterioration in trabecular bone microarchitecture and bone mass at this 10-week time point in T2DM rats. The current study builds upon that work by investigating histological and molecular mechanisms. Therefore, the chosen duration was based on prior evidence that this model reliably produces bone structural changes consistent with chronic diabetic bone disease.
Reference:
9. Mohsin S, Kaimala S, Sunny JJ, Adeghate E, Brown EM. Type 2 Diabetes Mellitus Increases the Risk to Hip Fracture in Postmenopausal Osteoporosis by Deteriorating the Trabecular Bone Microarchitecture and Bone Mass. J Diabetes Res. 2019;2019:3876957. doi:10.1155/2019/3876957.
Comment 2
2.Although the method of “high-calorie diet for 3 weeks + 2 STZ injections” to induce T2DM is commonly used, it only takes a few weeks and may not be enough to simulate the long-term effects of chronic T2DM on bone metabolism.Please verify its function by using miR-155 mimic/inhibitor or RhoA siRNA.
Response:
We completely agree with your comment. To simulate the long-term effects of chronic T2DM, both control and T2DM groups were maintained for 10 weeks after the onset of diabetes before sacrifice. In our earlier study, we verified significant changes in bone quality at this time point (Mohsin et al., 2019). The current study uses bones from the same cohort, as indicated in the manuscript (page 10, Materials and Methods, lines 375–376):
"The study utilized bones from three-month-old female Wistar rats (n = 16), which had been stored at -80°C following a previous investigation[9]."
To clarify, we rephrased line 397:
"After confirmation of diabetes, both the control and T2DM groups were maintained for an additional 10 weeks to allow chronic disease changes to develop before sacrifice."
Regarding the suggested functional experiments, these are already included as part of the study limitations and future directions. We have rephrased the text on page 13 (last para)-14 (first para) Line 512-516 to make it clearer:
"Future studies should include functional validation using miR-155 knockdown or overexpression (e.g., mimics/inhibitors) and RhoA siRNA, along with luciferase reporter assays and cytokine modulation, to elucidate these regulatory dynamics more conclusively."
Reference:
9. Mohsin S, Kaimala S, Sunny JJ, Adeghate E, Brown EM. Type 2 Diabetes Mellitus Increases the Risk to Hip Fracture in Postmenopausal Osteoporosis by Deteriorating the Trabecular Bone Microarchitecture and Bone Mass. J Diabetes Res. 2019;2019:3876957. doi:10.1155/2019/3876957.
Round 3
Reviewer 2 Report
Comments and Suggestions for Authors
The authors have answered my questions.